# Dual Demodulation of Temperature and Refractive Index Using Ring Core Fiber Based Mach-Zehnder Interferometer

**DOI:** 10.3390/mi12030258

**Published:** 2021-03-03

**Authors:** Weihao Yuan, Changyuan Yu

**Affiliations:** Photonic Research Centre, Department of Electronic and Information Engineering, The Hong Kong Polytechnic University, Hong Kong, China; weihao.yuan@connect.polyu.hk

**Keywords:** ring core fiber, Mach-Zehnder interferometer, temperature, refractive index

## Abstract

We report the ring core fiber spliced with single mode fiber and no core fiber which is used for temperature and refractive index (RI) sensing. The Mach-Zehnder interferometer (MZI) is formed with this kind of sandwich fiber structure and the maximum extinction ratio of the interference spectra reaches 27 dB with the free spectra range of 12 nm. The MZI fiber sensor is applied for temperature sensing with the sensitivity of 69 pm/°C and 0.051 dB/°C. The RI sensitivity reaches 182.07 dB/RIU and −31.44 nm/RIU with the RI ranging from 1.33 to 1.38. The RI value can be directly demodulated with the interference dip intensity which shows insensitivity to temperature. The demodulation of temperature can be achieved by using the linear equations between dip wavelength shift with the variation of temperature and RI.

## 1. Introduction

Optical fiber sensor has gained extensive attention in recent years due to its various advantages, such as low cost, ease of fabrication, compactness, good resistance to electromagnetic interference and chemical corrosion [1]. Diverse applications of fiber sensor have been researched and developed, for instance, temperature sensing [2,3,4], humidity sensing [5,6], refractive index (RI) sensing [7,8,9], strain sensing [10,11], curvature sensing [12,13], pH sensing [14,15], gas sensing [16,17,18] and so on. Temperature and RI are seemed as the important parameters because of their importance on environment monitoring, liquid quality inspection and food safety detection. Different fiber structures are studied for simultaneous measurement of temperature and RI, including Fiber Bragg Grating (FBG) [19,20], Mach-Zehnder interferometer (MZI) [21,22], Fabry-Perot interferometer (FPI) [23,24], side-polished fiber [25] and hybrid fiber structures [26,27]. The MZI fiber sensor is regarded as one of the most effective structures. The principle of MZI sensor for temperature sensing is mainly based on the changes of effective RI difference between core and cladding and that of interference length induced by thermal effect. The alteration of effective RI difference and interference length will lead to the wavelength shift of the interference spectra, which can be used for demodulation of temperature. The RI sensing can be realized by exciting the evanescent field on the fiber surface, on which surrounding materials can make the impact on the fiber cladding modes. This can give rise to the variation of light amplitude and phase which will further result in the change of interference spectra. 

In this paper, the ring core fiber (RCF) based MZI fiber sensor is proposed and experimentally demonstrated for dual demodulation of temperature and RI. The MZI sensor was fabricated by direct fusion splicing a piece of no core fiber (NCF) to a segment of RCF. Then the formed N-R fiber structure was connected to two pieces of single mode fibers (SMFs). The built SMF-NCF-RCF-SMF structure can be seemed as the miniatured MZI. Multiple fiber modes are stimulated in the RCF and combined into the SMF. The experimental results demonstrate the good interference effect of the MZI structure with the maximum extinction ratio of 27 dB and the free spectra range of 12 nm. The temperature sensing is realized by monitoring the wavelength shift of interference dip induced by heating and cooling effect and the RI sensing can be implemented due to the adjustment of cladding modes’ amplitude and phase induced by surrounding RI change. The RI value is expressed by refractive index unit (RIU). The interference dip intensity and dip wavelength shift display different responses to temperature and RI changes which makes it possible for dual demodulation of these two parameters. The variation of temperature can result in the wavelength shift with the slope of 69 pm/°C and negligible change of dip intensity in limited temperature ranges. The alteration of RI values gives rise to simultaneous wavelength shift of −31.44 nm/RIU and dip intensity change of 182.07 dB/RIU. Thus, the RI can be demodulated by the change of interference intensity and temperature sensing can be achieved through the linear relationship between dip wavelength with the value of temperature and RI.

## 2. Experiment and Analysis

The MZI was fabricated with four segments which consisted of double commercial single-mode fibers (SMF-28, Corning, NY, USA), no core fiber and ring core fiber. The NCF and RCF were purchased from YOFC and the diameters of SMF, NCF and RCF are the same at 125 μm which enables the direct fusion splicing without diameter mismatching. The lengths of applied NCF and RCF are 1mm and 20 mm, respectively. As can be seen from Figure 1a, the cross section of RCF can be divided into three parts: fiber center (FC), ring core (RC) and cladding area. The radii of FC, RC and cladding are 5.45 μm, 9.35 μm and 62.5 μm, separately. Figure 1b displays the RI distribution along the fiber diameter (yellow dash line) and the RI of RC is 0.0134 larger than that of FC and cladding with the value of 1.44402.

Figure 2 depicts the schematic diagram of the MZI with the SMF-NCF-RCF-SMF structure. The blue parts show the SMFs with the diameter of 125 μm and core diameter of ~8.5 μm. The grey part represents the NCF with the diameter of 125 μm, which has the same RI value with that of SMF cladding. The yellow part depicts the RCF and the details are shown in Figure 1. The lengths of applied NCF and RCF are 1 mm and 20 mm, respectively. Broadband Source (BBS) with the bandwidth from 1470 nm to 1670 nm is used as the light source. The light transmitted in SMF will firstly be coupled into NCF and then be transmitted to RCF in which multiple fiber modes will be excited including FC mode, RC mode and cladding modes. Different fiber modes will interfere and be coupled in SMF again at the splicing node of RCF and SMF. The MZI sensor is placed in heating oven for temperature sensing and the liquids with different sugar concentration are applied for RI sensing. The RI values of different liquids are measured and labeled using commercial saccharimeter. The optical spectrum analyzer (OSA, AQ6370D, YOKOGAWA, Japan) with the measuring range of 600-1700 nm is utilized for spectra recording.

To study the effect of NCF on the MZI structure, the Rsoft simulation work was done and the results were shown in Figure 3. The modeling of fiber structures with SMF-RCF-SMF and SMF-NCF-RCF-SMF was conducted with the simulation tool of BeamPROP. For SMF, the diameters of core and cladding were set to be 8.5 μm and 125 μm with the RI values of 1.4504 and 1.4447, respectively. The lengths of NCF and RCF were built as 1 mm and 20 mm. The RCF were modeled under the measurement results shown in Figure 1. The wavelength of 1550 nm was chosen as the free space wavelength and the grid sizes on different axes of X, Y and Z were set to be 0.2 μm. As can be seen from Figure 3a, the light is coupled from SMF to RCF and the mismatched interface between SMF and RCF gives rise to the energy redistribution which excites multiple modes in RCF including the FC mode, RC mode and cladding modes. The light energy is mainly coupled in FC mode and RC mode which induces the relatively weak cladding modes. By applying the NCF, as shown in Figure 3b, the cladding modes are enhanced and RC mode becomes weaker which means that the NCF results in the redistribution of light energy among FC mode, RC mode and cladding modes. More cladding modes are excited in RCF which can significantly enhance RI sensitivity due to the strong interaction of cladding modes and surrounding materials. The Rsoft simulation results indicate that the MZI based on SMF-NCF-RCF-SMF can potentially serve as preferable sensor structure on RI sensing compared to that based on SMF-RCF-SMF structure.

The interference spectrum of the MZI sensor based on SMF-NCF-RCF-SMF structure is displayed in Figure 4 (inset). No distinct interference pattern can be recorded if the NCF is removed from the fiber structure. The spectrum is recorded from 1500 nm to 1600 nm using OSA with the resolution of 0.02 nm. The maximum extinction ratio (ER) is measured to be 27 dB and free spectral range (FSR) is 12 nm. The large ER and FSR enable the demodulation of environmental parameters by utilizing both dip wavelength and intensity. The fast Fourier transform (FFT) is conducted and the FFT spectrum is shown in Figure 4. Except for the fundamental mode, multiple high-order modes are also excited which is in correspondence with the simulation results depicted in Figure 3b. The relationship between interference spectra variation and the external environment change can be explained by the interference equations below: (1)$$I=I_{1}+\underset{k}{\sum }I_{2}^{k}+\underset{m}{\sum }2\cdot \sqrt{I_{1}\cdot I_{2}^{k}}cos[\frac{2\pi }{\lambda }\cdot \left(n_{eff}^{1}-n_{eff}^{2,k}\right)\cdot L]$$

I, $I_{1}$ represent the intensity of output light and RC mode. $I_{2}^{k}$ means the intensity of FC mode and multiple cladding modes. $n_{eff}^{1}$ represents the effective RI of the RC mode and $n_{eff}^{2,k}$ shows that of the FC mode and cladding modes. *L* is the interference length which can be seemed as the length of the RCF. The condition of destructive interference can be explained with the following equations:(2)$$2\pi \cdot \left(n_{eff}^{1}-n_{eff}^{2,k}\right)\cdot L/\lambda =\left(2m+1\right)\pi$$
(3)$$\lambda =\frac{2\Delta n_{\mathrm{eff}}L}{2m+1}$$

$\Delta n_{\mathrm{eff}}$ means the RI difference between the RC with the FC and the cladding. λ in Equation (3) means the wavelength of the interference dips.

Figure 5 depicts the sensing performance of the RCF based MZI at different temperatures ranging from 35 °C to 70 °C with the interval of 5 °C. The experiment was conducted in the air with surrounding RI of 1.0. As displayed in Figure 5a, the wavelength of the interference dip shows continuous shift with temperature change which can be attributed to the alteration of effective RI difference ($\Delta n_{\mathrm{eff}}$) and the interference length (*L*) induced by thermal effect. The thermal-optic coefficient of Ge-doped silica core is larger than that of fused silica cladding which means that $\Delta n_{\mathrm{eff}}$ will increase with the rising of the temperature and the red shift of dip wavelength can be observed [28]. In addition, the rising and declining of temperature will give rise to the expansion and shrink of the fiber length and further lead to the red and blue shift of the dip wavelength. This can be explained by Equation (3). The mathematical statistics between the temperature and the dip wavelength are implemented and the result is displayed in Figure 5b. The red line represents the linear fit in the case of temperature increasing and blue line shows that of temperature decreasing. The relationship of temperature and dip wavelength shows good linearity in the process of both temperatures rising and declining and the adjusted R square values (R^2^) are both 0.99. The slope of temperature increasing is calculated to be 68 pm/°C and 69 pm/°C slope is achieved when temperature decreases. The results are better than that reported in Reference [29] which can be attributed to the different ring core size of the applied fiber. 

The statistics of temperature and interference intensity are also conducted which is depicted in Figure 6. The dip intensity shows very slight alteration as the temperature changes. The linear fit gives the slope of 0.051 dB/°C which means that the intensity of interference dip at around 1508 nm is insensitive to thermal effect.

The MZI can also be used for RI sensing because the excited cladding modes induced by NCF are sensitive to surrounding RI and the results are shown in Figure 7. Unlike the FC mode and RC mode which transmit around the center of RCF, cladding modes will be reflected on the interface of fiber cladding and air. Thus, cladding modes can be sensitive to the variation of surrounding materials which can easily tune both the amplitude and phase of the cladding modes. The interference spectra will display corresponding alteration according to the change of external environment. Figure 7a presents the sensing results with different values of RI ranging from 1.33 to 1.38 with the interval of 0.01 at room temperature (~25 °C). With the step increasing of surrounding RI, the interference spectra show blue shift which can be explained by Equation (3). In addition, the change of external RI values can also lead to the amplitude change of cladding modes which gives rise to the alteration of interference intensity as shown in Equation (1). Figure 7b displays the relationship between RI and the dip intensity. The experiment is repeated and the error bars are signed as black triangle. The RI sensitivity is calculated to be 182.07 dB/RIU with the linearity of 0.98. Compared to the dip intensity change induced by thermal effect, which is 0.051 dB/°C shown in Figure 6, that influenced by RI is obviously much more remarkable. This makes it possible for direct demodulation of RI with the dip intensity by ignoring the weak influence generated by thermal effect. Figure 8 depicts the linear relation between RI and dip wavelength and the slope is calculated to be −31.44 nm/RIU. The temperature hence can also be demodulated by using the linear equations between wavelength shift with the variation of RI and temperature:
(4)$$\Delta \lambda \;=0.069\times \;\Delta T\;+\;\left(-31.44\right)\;\times \;\Delta RI$$
(5)$$\Delta T\;=\;\frac{1}{0.069}\;\left(\Delta \lambda \;+\;31.44\times \;\Delta RI\right)$$
where Δλ means the wavelength shift, ΔT is the change of temperature and ΔRI represents that of RI.

## 3. Discussion

Although the influence of thermal effect on interference dip intensity is slight (0.051 dB/°C) in the temperature range from 35 °C to 70 °C based on our measurement, large temperature variation can still result in a nonnegligible impact on dip intensity which will lead to a relatively large error on the demodulation of RI. Thus, in our experiment, the dual demodulation of temperature and RI will be confined in a finite temperature range (within 20 °C variation) to make sure the measurement error be controlled within reasonable limits. Future work can be done to seek solution of such demodulation method that enables sensor application within large measurement range, such as optimizing fiber structure or utilizing material assistance. 

## 4. Conclusions

In conclusion, the MZI fiber sensor based on SMF-NCF-RCF-SMF structure is successfully achieved. The experiments on temperature and RI sensing are conducted and dual demodulation of temperature and RI can be realized with the temperature variation under 20 °C and RI resolution of 0.01 ranging from 1.33 to 1.38. The maximum sensitivity of temperature sensing is 69 pm/°C and that of RI sensing reaches 182.07 dB/RIU and −31.44 nm/RIU with the intensity and wavelength demodulation methods, respectively. The proposed MZI sensor has advantages of cost-effective source, ease of fabrication and high sensitivity for both temperature and RI sensing. In addition, based on the cladding modes modulation, the MZI sensor reveals good potential for diverse applications, such as relative humidity sensing and pH sensing assisted with chemical materials. Future works will be conducted to extend its applications on variously environmental monitoring. 

## Figures and Tables

**Figure 1 micromachines-12-00258-f001:**
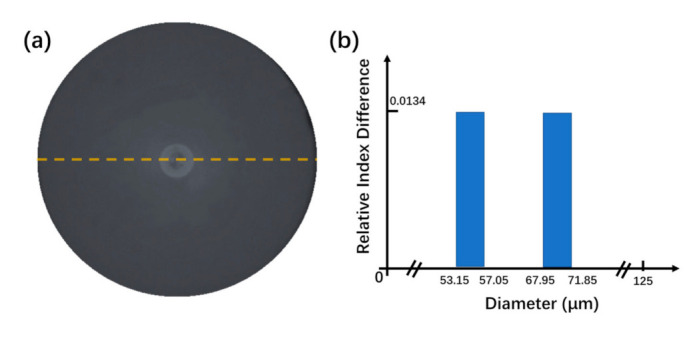
The microscope images of RCF (**a**) and the RI distribution along the diameter (dash line) (**b**).

**Figure 2 micromachines-12-00258-f002:**
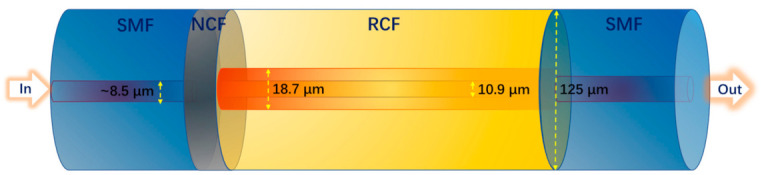
The schematic diagram of the MZI with the SMF-NCF-RCF-SMF structure.

**Figure 3 micromachines-12-00258-f003:**
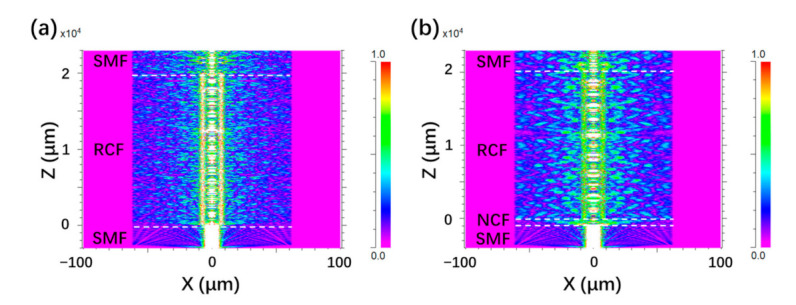
The Rsoft simulation results of SMF-RCF-SMF structure (**a**) and SMF-NCF-RCF-SMF structure (**b**).

**Figure 4 micromachines-12-00258-f004:**
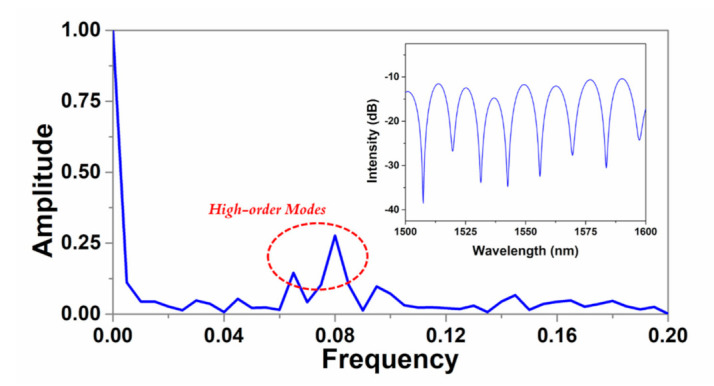
The interference spectrum of the MZI (inset) and the FFT spectrum.

**Figure 5 micromachines-12-00258-f005:**
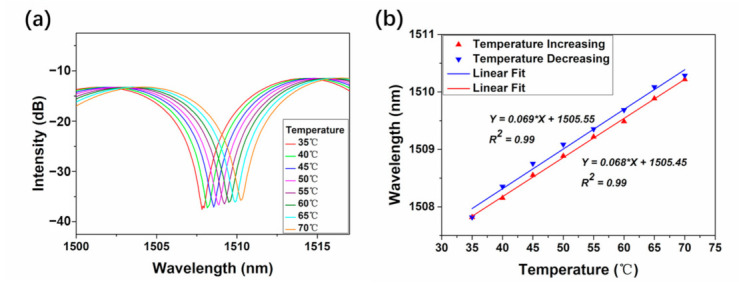
The MZI interference spectra at different temperatures ranging from 35 °C to 70 °C with the interval of 5 °C (**a**) and the mathematical statistics of temperature and dip wavelength with procedure of temperature increasing and decreasing (**b**).

**Figure 6 micromachines-12-00258-f006:**
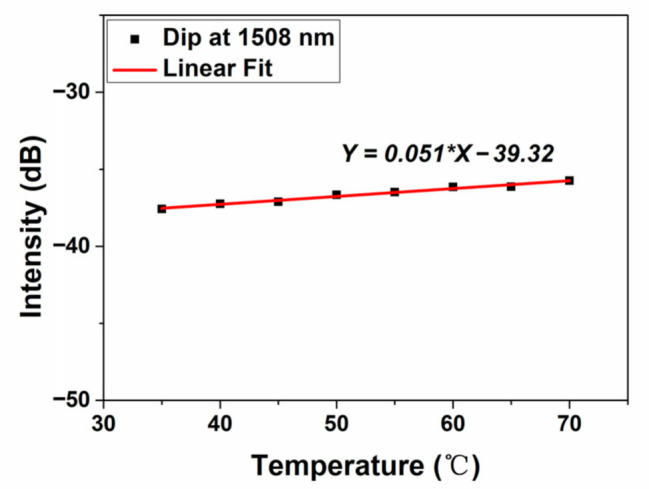
The relationship between interference dip intensity and temperature change at the dip wavelength of around 1508 nm.

**Figure 7 micromachines-12-00258-f007:**
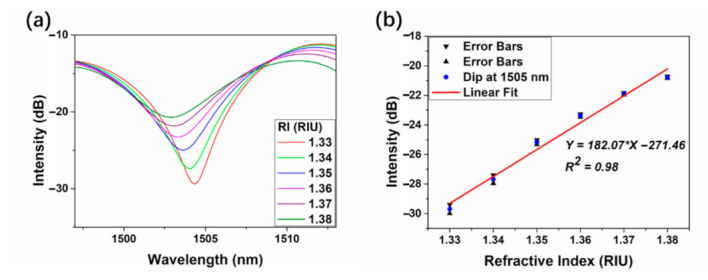
The MZI interference spectra with different RIs ranging from 1.33 to 1.38 with the interval of 0.01 (**a**) and the relationship between RI and the intensity of interference dips (**b**).

**Figure 8 micromachines-12-00258-f008:**
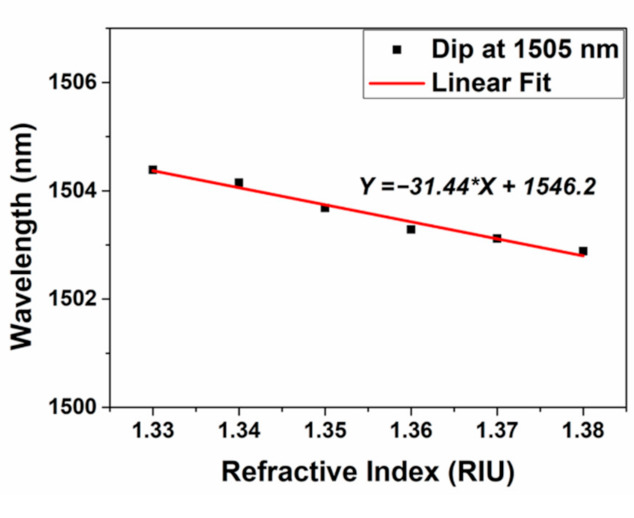
The mathematical statistics of RI and wavelength shift at the dip of around 1505 nm.

## Data Availability

Not applicable.

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
