# Peer review of "Dual Demodulation of Temperature and Refractive Index Using Ring Core Fiber Based Mach-Zehnder Interferometer"

_micromachines, 2021, doi:10.3390/mi12030258_

Round 1

Reviewer 1 Report

The manuscript describes a multi-section optical fiber sensor for temperature and refractive index sensing. The overall approach seems sound, though not entirely novel.

The reviewer has to point out that neither the title nor the conclusion was fully justified by the results presented in the manuscript.

Above all, the reviewer does not think that temperature and index were fully demodulated. They are measured and represented simultaneously and there seems no definite way to separate them.

If the authors would like to claim dual-mode operation, they should specify the range for both parameters (temp and index) in which the sensor can stay reliable.

Practically, 0.05 dB/C or 1 dB/20C is a big change which cannot be just deemed negligible.

Was there any attempt to get the output while the temperature and index were simultaneously being varied? If not, please explain why not.

Please specify

(1) The refractive index at which the results in Fig. 5 and 6 were taken,

(2) The temperature at which the results in Fig. 7 and 8 were taken.

The authors claimed that their results are better than that reported in [29]. They should explain why theirs fared better.

The use of undefined acronyms in the title should be avoided.

The fiber dimension information in the first paragraph of Section 2 should also appear in Fig. 2 for easier reading.

In Fig. 3, please do not compress the plots in the z-direction. Much of the details were lost.

"Sugar degree" > "Sugar concentration"

FSR should be specified in frequency, not wavelength.

Most legend texts in Fig. 5, 6, 7, 8 are illegibly small. Please change the font size.

The reviewer thinks that the manuscript can be further improved.

Reviewer 2 Report

The paper describes the design and testing of a sensor based on a MZI realized using only the splicing of different fiber types. The English language is appropriate and information is well presented, overall.

There are, however, several issues that require attention.

  1. Page 3, second paragraph: grid size information is insufficient. How was the cell size determined? Is the 0.5 µm in the X and Y, and 1 µm in the Z directions appropriate for the 1.55 µm wavelength of the source? Would making the grid cell smaller improve the simulation results?
  2. Page 3, second paragraph: what about sensor stability? Your design indicates that the sensor must be straight in order to function properly. But what happens if it is bent? How will it affect the results, and by how much?
  3. Figure 4: could a longer acquisition time improve the amount of information contained in the spectrum plot? In other words: would it improve measurements if the plot had a bit more data?
  4. Figure 5 and 7: are these plots for the sensor with the NCF, or without it? You mentioned the NCF improves RI sensitivity, but what does it do for the temperature sensitivity? If it is overall better with the NCF, why did you describe the device without the NCF as well? (i assume it was the natural progression of development of the device, but I fail to see the necessity of introducing this much information about the device without NCF).

Reviewer 3 Report

In this manuscript entitled "Dual Demodulation of Temperature and Refractive Index Using SMF-NCF-RCF-SMF Fiber Structure", the authors fabricated a fiber structure with its operation wavelength around 1505 nm. The authors claimed that the proposed device can be used as a refractometric sensor and temperature sensor. The manuscript seems to have a good structure, but because of some technical concerns, I think major revision is needed before it can be published in Micromachines. Please see my comments below:

  1. The authors claimed that they fabricated an MZI based on an SMF-NCF-RCF-SMF fiber structure, but how does this fiber structure work as an MZI is not demonstrated. I cannot agree this is something that obvious enough for all readers to understand. It would be more convincing if the authors can provide such evidence.
  2. The English language used in the manuscript is not scientifically accurate. There are also typos such as “extinction ration” (even in the abstract) and grammar mistakes throughout the whole manuscript. Sometimes I need to guess what the authors wanted to convey. Some examples:
  3. Page 1, Line 32-33, “between core with cladding and interference length …” => I guess it should be “between core and cladding and that of interference length… ”?
  4. Page 2, Line 50, what is “The interference depth”?
  5. Page 2, Line 56, “… temperature can be achieved…” => I guess “sensing” is missing after “temperature”?

...

  1. I am confused by Page 3, Line 79 “Broadband Source (BBS) with the bandwidth from 1470 nm to 1670 nm is used as the laser source.” Were the authors using a broadband light source or a laser? In my understanding, lasers should be typical narrowband light sources. How can a BBS be used as a laser source? Please correct me if I am wrong. From the context, I guess the authors were not using a laser. But it was really confusing, especially when “laser” was used multiple times.
  2. It is not easy to distinguish the fiber segments in Figure 3 a) and b). Maybe adding some assistive lines to show the fiber segment boundaries can show the impact of NCF better.
  3. I suggest swap Figure 4 and its inset. Usually, we look at the main figure first, then its inset. But the authors did the opposite way.
  4. Figure 5(b), why there is a small difference in the dip wavelength at each temperature between “Temperature Increasing” and “Temperature Decreasing”? From the plot, we can see that the data points in “Temperature Decreasing” are very close to the neighboring data points in “Temperature Increasing” which are 5 degrees away. For example, the blue triangle at 45C is so close to the red triangle at 50C in the spectrum. Given a dip wavelength, how can you be sure if it is at 45C or 50C? Besides, the term “dip at 1505 nm” is not accurate, since the dip is not at 1505 nm.
  5. Why does the dip intensity change with RI? Why does the dip become wider as the RI increases?

Round 2

Reviewer 1 Report

I think my suggestions were mostly addressed.

But the author's response to Point 3 was not sufficient.

The reviewer wanted the authors to specify

  • The temperature at which the data in Fig. 7 and 8 were taken.
  • The refractive index at which the data in Fig. 5 and 6 were taken.

These requests were not taken care of.

Other than that, the reviewer could not find any issue.

Author Response

Point 3: Please specify

(1) The refractive index at which the results in Fig. 5 and 6 were taken,

(2) The temperature at which the results in Fig. 7 and 8 were taken.

Response 3: We thank the reviewer’s comment and suggestion.

We have revised the manuscript to specify the information which can be found in Page 6, Line 144-146 and Page 8, Line 185-187:

Figure 5 depicts the sensing performance of the RCF based MZI at different temperatures ranging from 35 C to 70 C with the interval of 5 C. The experiment was conducted in the air with surrounding RI of 1.0.

Figure 7a presents the sensing results with different values of RI ranging from 1.33 to 1.38 with the interval of 0.01 at room temperature (~25 C).

Reviewer 3 Report

The authors have properly addressed my concerns. Therefore I recommend this manuscript to be published on Micromachines 

Author Response

We thank and appreciate the reviewer's comments and suggestions for the manuscript revision.